# In Medical Claims Data, Enhancing Predictive Performance for Major Adverse Cardiovascular Events Using Cross Attention

Yuhei Fujioka*
Human Health Sciences, Kyoto University Graduate
School of Medicine
Cancerscan Inc.
Tokyo, Japan
y.fujioka@cancerscan.jp

Tatsuyoshi Ikenoue†
Data Science and AI Innovation Research Promotion
Center, Shiga University
Shiga, Japan
tatsuyoshi-ikenoue@biwako.shiga-u.ac.jp

Daitaro Misawa*
Human Health Sciences, Kyoto University Graduate
School of Medicine
Cancerscan Inc.
Tokyo, Japan
misawa@cancerscan.jp

Shingo Fukuma
Human Health Sciences, Kyoto University Graduate
School of Medicine
Kyoto, Japan
fukuma.shingo.3m@kyoto-u.ac.jp

## ABSTRACT

Medical claims data comprise the financial details, including the expenses and billing information, as well as the clinical information, such as the diagnoses and treatments, of patients visiting medical facilities. Recently, it has been acknowledged that large databases can be constructed from medical claims data for medical research purposes. However, the clinical information within these datasets is often medically unstructured, limiting its application in comprehensive analyses. This study enhances predictive model performance for major adverse cardiovascular events (MACE), a leading cause of death worldwide. Models that predict MACE are crucial to clinical practice guidelines. We utilize a cross-attention mechanism to develop a method that effectively weights the relationships between diagnoses and treatments. Effectively representing the clinical information contained in medical claims data, this approach generates more representative features for predicting MACE. The ROC-AUC score of our proposed cross-attention-based model was 0.7720, higher than other benchmark models including the conventional atherosclerotic cardiovascular disease model, the light gradient boosting machine, and a self-attention-based model. These results indicate that integrating the clinical structure of medical claims data using a cross-attention mechanism significantly enhances the performance of predictive models.

## CCS CONCEPTS

• **Applied computing** → **Health informatics**; • **Computing methodologies** → *Machine learning approaches.*

---

*Both authors contributed equally to the paper

†Also with Human Health Sciences, Kyoto University Graduate School of Medicine.

---

## KEYWORDS

medical claims data, deep learning, cross-attention, major adverse cardiovascular events, healthcare

**ACM Reference Format:**

Yuhei Fujioka, Daitaro Misawa, Tatsuyoshi Ikenoue, and Shingo Fukuma. 2024. In Medical Claims Data, Enhancing Predictive Performance for Major Adverse Cardiovascular Events Using Cross Attention. In *Proceedings of Make sure to enter the correct conference title from your rights confirmation email (AIDSH-KDD '24).* ACM, New York, NY, USA, 13 pages. https://doi.org/XXXXXXX.XXXXXXX

## 1 INTRODUCTION

### 1.1 Background

In recent years, the use of medical claims data and electronic health records (EHR) has been increasingly recognized as crucial in developing disease prediction models [23, 29]. These datasets comprehensively record medical actions for each patient, including diagnoses, medical procedures, and prescriptions. Since this information provides a comprehensive view of patients' health statuses, it is expected that interrelating diagnoses and treatments (defined as medical procedures and prescriptions in this study) will enhance disease prediction performance. However, only a few researchers are effectively utilizing this information to develop disease prediction models. For example, while Rupp et al. [24] linked diagnoses and treatments in a simple and arbitrary manner, the relationship between them was not adequately considered, only correlating treatments to a diagnosis (many-to-one relationship). This is important as the relationships between diagnoses and treatments can often be many-to-many. For instance, for the diagnoses of hypertension and chronic heart failure, the treatments could be Captopril for the former or Sacubitril-Valsartan for both. While it is conceivable to manually link diagnoses and treatments, this approach is limited by the availability and accuracy of domain knowledge, it introduces the risk of subjectivity due to human judgment, and is time-consuming and inefficient. When predicting major adverse cardiovascular events (MACE), health checkup data, including laboratory and self-reported data, are used extensively [4, 5, 16, 25]. MACE include hospitalizations or deaths that occur due to major

cardiovascular diseases such as acute myocardial infarction, cerebrovascular disease, heart failure, and peripheral arterial disease. MACE are not only a leading cause of death worldwide [15, 19] but also pose a high risk of disability and premature death [26, 30], and create a significant social burden. Therefore, preventing MACE is an urgent issue. Identifying high-risk individuals and appropriately managing their risk factors is considered one of the effective approaches to its prevention [16]. In their clinical practice guidelines, various countries have adopted statistical models to predict MACE [4, 5, 16, 25], calculate the risks, and implement interventions based on the risk levels. However, these models do not fully utilize the clinical information available in medical claims data.

## 1.2 Task Definition

This study focuses on the prediction of MACE. The input data for our models include health checkup data and medical claims data. A model's output indicates the probability of a patient experiencing hospitalization or death (i.e., the event) due to MACE, which are defined as any of the diseases listed in Table 1. The diseases are identified with the International Classification of Diseases 10th Revision (ICD-10) codes [17], which are used for generating and analyzing statistics related to causes of death and diseases. Since the incidence of MACE is very low (Table 2), our classification task is challenging.

**Table 1: ICD-10 codes for cardiovascular diseases**

| Disease | ICD-10 |
| --- | --- |
| AMI | I20 – I25 |
| CEREBRO | I60 – I69 |
| HF | I50 |
| PAD | I70 |

The expanded forms of the diseases listed in the table are acute myocardial infarction (AMI), cerebrovascular disease (CEREBRO), heart failure (HF), and peripheral arterial disease (PAD)

## 1.3 Challenges

Linking diagnoses to treatments is crucial as it can provide a deeper understanding of the clinical information contained within medical claims data. However, despite being based on a series of actions performed on each patient, these data lack a clear medical structure. In many countries, linking diagnoses to their treatments in medical claims data is difficult [12], primarily because medical claims data are designed for healthcare provider reimbursement. To address this issue, our study utilizes the cross-attention mechanism to complementarily weight the relationship between diagnoses and treatments, allowing for appropriate matching between them, even when their relationship is many-to-many. Thus, we enhance the model's performance in predicting MACE by effectively utilizing the clinical information contained in medical claims data.

## 1.4 Contributions

In this study, we develop a deep learning model that includes a cross-attention mechanism as a crucial part of the transformer architecture. The objective is to predict MACE by integrating health checkup data and medical claims data. Our findings highlight that the model's performance is significantly improved by the combination of these two data types and the cross-attention mechanism, compared to using only health checkup data. We particularly emphasize that, unlike models in recent studies that rely solely on the self-attention mechanism and learn from a many-to-one relationship between diagnosis and treatment (not considering a many-to-many relationship), our proposed model leverages the cross-attention mechanism to enable learning from a many-to-many relationship. We also show that our approach of integrating a cross-attention mechanism with medical claims data effectively enhances the performance of MACE prediction.

## 2 RELATED WORK

### 2.1 MACE Prediction Models

Pooled Cohort Equations [4] were developed using health checkup data (including laboratory and self-reported data) from African American and white individuals aged 40–79 to predict the 10-year risk of atherosclerotic cardiovascular disease (ASCVD). These equations have been adopted in American clinical guidelines [4] and are widely cited [6, 8, 25]. In contrast, the Suita score model, based on data from Japanese urban residents aged 30–79, estimates the 10-year risk of coronary heart disease and is included in the Japan Atherosclerosis Society's clinical guidelines.

### 2.2 Disease Prediction Utilizing Health Checkup Data and Clinical Information

High prediction performance for both the onset and progression of disease has been achieved using the health checkup data and clinical information contained in medical claims data and EHR [9, 27]. Importantly, these studies did not use all the diagnosis and prescribed medication codes recorded in the EHR, only employing the clinical information related to the specific outcomes they aimed to predict. However, they reported significant improvements in the model's ROC-AUC.

### 2.3 Self-Attention in Clinical Information

Medical claims data and EHR provide a wealth of information related to medical care, and numerous studies on predicting various diseases using these data are being conducted. Among these, models that utilize EHR data and incorporate a self-attention mechanism have achieved positive results and have garnered much attention [13, 21, 22, 24]. Including both diagnosis and treatment in a patient's sequence improves disease prediction performance of a model based on a self-attention mechanism [21]. However, as the length of a patient's sequence increases, the computational resources required for machine learning also increase. As a result, the number of patients that can be applied to training using the self-attention mechanism becomes limited. To address this, ExBEHRT [24] links each treatment to a diagnosis, thereby improving prediction performance and reducing hardware requirements. However, this approach overlooks the many-to-many relationship between diagnoses and treatments, which is a scenario that has not been addressed. Manually linking diagnoses and treatments presents a challenge since their correlation in medical claims data and EHR is not always clear,

making accurate determination of the relationships difficult and time-consuming. Furthermore, researchers could introduce bias in the linking process through their subjectivity

## 2.4 Examples of the Application of Cross-Attention Mechanisms

Cross-attention is one of the mechanisms used in the decoder of the transformer architecture. Unlike self-attention, cross-attention processes two distinct inputs and calculates the relevance of each element from one input to all elements from the other input. Using this calculation, it determines which elements are most closely related and aggregates the calculated relevance (i.e., it weights the relationship). This technique has achieved positive results in supervised learning. For instance, in the field of drug discovery, the accurate prediction of which proteins interact with which compounds has led to the development of new medications. The correlation between the amino acids that make up proteins and the elements that make up compounds is also a many-to-many relationship, and handling this relationship with deep learning models has been a challenge. Addressing this issue, studies utilizing cross-attention to learn the relationships between proteins and compounds as inputs have shown superior predictive outcomes [11, 18].

## 3 METHOD

## 3.1 Model

*3.1.1 Our Proposed Model (Our CA).* This chapter first outlines the general structure of the model architecture we propose. To clarify the complex process, we provide a detailed explanation of how the input features were handled within the embedding block. Lastly, we elaborate on the processing of the cross-attention mechanism, which is a key component of our proposed model.

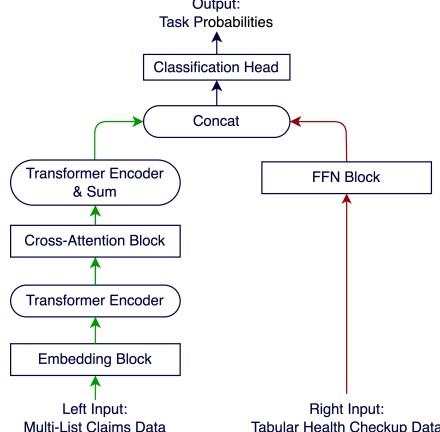

**Figure 1: Proposed model architecture. The "sum" reduces the length dimension. See Figure A.3 for detailed information.**

- **Model Architecture**: Our proposed model architecture is depicted in Figure 1, and the detailed architecture of each specific block is shown in Figure A.3. Two types of input were used for this

model: 1) tabular data pertaining to health checkups; and 2) multi-list data obtained from one year of medical claims data, including diagnoses, medical procedures, and prescriptions, organized into 12 monthly lists. The tabular data were first transformed using a feed-forward network (FFN) block, after which the multi-list data were processed through an embedding block that integrated the medical procedures and prescriptions into the treatment data. Following this, the diagnoses and treatments were separately transformed for each month using their respective transformer encoders [28]. This was followed by the cross-attention block to create a vector that represents monthly clinical information, integrating the diagnoses and treatments by considering their interrelations (this process is explained further in Cross-Attention Block on page 3). Each month's data were then combined into a single tensor. By concatenating the tensors of each month, we represented 12 months of medical claims data as a tensor with a length of 12. The transformed results for both the tabular and medical claims data were then concatenated and passed through a classification head at the top of the architecture, which outputs the probabilities. The FFN block comprised a linear layer, batch normalization, a ReLU activation function, and dropout, while the embedding block consisted of an embedding layer, a sum layer, and a concat layer. Lastly, the classification head comprised a linear layer, L2 normalization, and a scale layer [20].

- **Embedding Block**: The multi-list data input was derived by breaking down the monthly medical claims data into tokens: the diagnoses, medical procedures, and prescriptions. To obtain more meaningful medical information for the predictions, these tokens were further broken down into sub-tokens, converting each diagnosis token into five hierarchical sub-tokens according to the World Health Organization ICD-10 Instruction Manual (§2.4.2–2.4.6). In addition, the prescriptions (prescribed medications) were divided according to their effects, administration routes and ingredients, and medication shape. Diagnosis sub-tokens were embedded and summed to create a unified diagnosis vector. Similarly, the vectors that represented medication effects, administration routes and ingredients, and medication shape were embedded, summed, and consolidated into a single prescription vector. By passing these three medical vectors through the sum layer, they were concatenated in the concat layer and employed as the output of the embedding block (refer to Figure A.4).

- **Cross-Attention Block**: Our proposed model uses cross-attention to complementarily weight both diagnoses and treatments, effectively establishing a strong connection between treatments closely related to the diagnosis. As shown in Figure 2, we provide a detailed explanation of how the model handles the relationship between diagnoses and treatments using cross-attention. The steps can be summarized as follows.

- Step 1: Calculate attention weights using the diagnosis and treatment data.
- Step 2: Apply attention weights to the treatments to generate a weighted treatment approach.
- Step 3: Enrich the diagnosis vector by integrating this weighted treatment approach to provide an output that contains more information about the treatments in relation to the diagnoses.
- Step 4: Apply layer normalization and input the results into an FFN, then reduce the length dimension with "sum".

Details of the parameters are presented in Table B.4.

### 3.1.2 Benchmark Models.

- **Pooled Cohort Equations**: The ASCVD model is a Cox proportional hazards model designed to estimate the 10-year risk of ASCVD events [4]. Following the guidelines [4], we applied the model developed for the white population to our dataset primarily composed of Japanese individuals.
- **Light Gradient Boosting Machine**: The light gradient boosting machine (LGBM) [10] is a well-known gradient boosting framework appreciated for its ability to learn from large-scale datasets quickly and efficiently.
- **Self-Attention Based Model (Our SA)**: Many recent studies have achieved excellent results with deep learning models that use the self-attention mechanism. To verify whether our proposed model, which leverages the cross-attention mechanism, can perform well compared to these advanced models, we developed a model that uses the self-attention mechanism for comparison.

The models mentioned in [13, 21, 22, 24] rely on detailed temporal information on diagnoses and treatments. Our medical claims data records diagnosis and treatment information on a monthly basis, lacking the detailed temporal information necessary to train the models in these previous studies. Therefore, it was difficult to apply these models to our dataset, and we did not adopt them as benchmarks in our study.

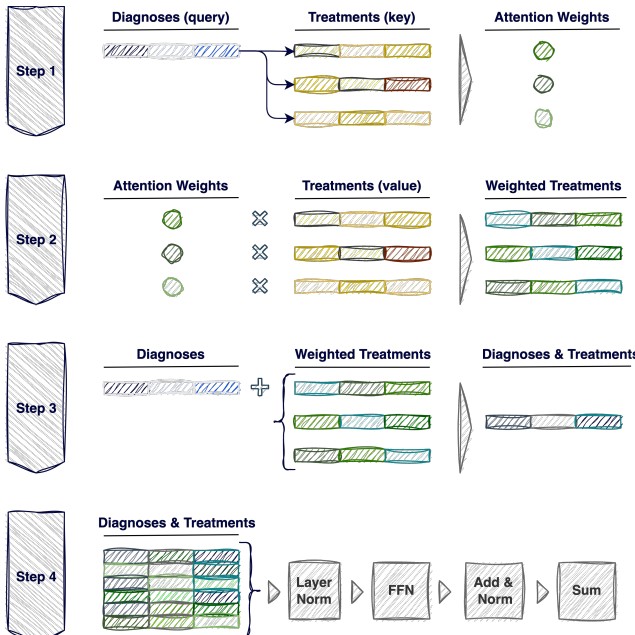

**Figure 2: Cross-attention block: In this approach, "diagnoses" serve as the query, while "treatments" are designated as both the key and the value for executing cross-attention. The application of attention weights integrates the weighted treatments into the vector of diagnoses and treatments. Following a process similar to the steps in a transformer encoder, layer normalization and feed-forward networks are utilized.**

## 3.2 Input Feature

Our models utilized two types of data as input features: 1) tabular health checkup data obtained from health checkup records; and 2) a list of claims data obtained from medical claims records. In addition, we prepared the input features specifically for the LGBM. Medical claims data were transformed into tabular format, which are referred to as tabular claims data. The LGBM used tabular health checkup data and tabular claims data as input features. This section describes the methods used to create these features.

### 3.2.1 Tabular Health Checkup data.
To create the tabular data, we extracted eight features from the health checkup data (see Table C.5). These features were then used as inputs in the ASCVD model developed by the American Heart Association and the American College of Cardiology.

### 3.2.2 Multi-List Claims data.
Monthly medical claims data were formatted as a list, including the codes for both the diagnoses and treatments. The treatment codes comprised both the medical procedure and prescription codes. To prevent data redundancy, duplicated codes were removed. For example, if the same medication was prescribed multiple times within a month, the redundant codes were removed. This decreased the size of the list and reduced the training time for the model. We also transformed the diagnosis, medical procedure, and prescription codes into sub-tokens using a more comprehensive medical classification (see Table D.6). For instance, the ICD-10 code "E112" can be converted into a list representing five medical categories: 1) E, referring to "endocrine, nutritional, and metabolic diseases"; 2) E10–E14, denoting "diabetes mellitus"; 3) E11, indicating "type 2 diabetes mellitus"; 4) E112, referring to "type 2 diabetes mellitus with renal complications"; and 5) <PAD>, denoting "not applicable". Thus, the medical codes used in monthly claims data can be organized into a set of list. These diagnosis, medical procedure, and prescription codes contained in monthly medical claims data are included in list format, but we broke down the these medical codes into meaningful sub-tokens and used them as input features. For instance, the diagnosis data for a given month in medical claims data were represented as a set of list, as follows:

$$\text{List}\left[\prod_{i=1}^{5}\text{ICD-10}_i\right] = \left\{[A_1,\ldots,A_n] \mid A_n \in \prod_{i=1}^{5}\text{ICD-10}_i\right\}$$

*where*

$$\prod_{i=1}^{5}\text{ICD-10}_i = \{(a_1, a_2, a_3, a_4, a_5) \mid a_i \in \text{ICD-10}_i\}$$

$\text{ICD-10}_i$ : ith category of ICD-10 code

The lists of diagnosis, medical procedure, and prescription codes from the monthly medical claims data were then processed and combined to create a comprehensive list spanning 12 months. This became the multi-list claims data, which served as the input for the self and cross-attention mechanism-based models.

### 3.2.3 Tabular Claims data.
Tabular claims data were generated using the diagnosis, medical procedure, and prescription codes included in the 12 months of medical claims data. This dataset consisted of nearly 7,000 different codes divided into columns, with

each code assigned a value that indicated whether it was present in (1) or absent from (0) the medical claims data.

## 3.3 Training

The AdamW optimizer [14] was used to train our models. To address the imbalance between the positive and negative examples in the task labels, class-balanced focal loss [3] was employed as the loss function. Optuna [1] was employed to tune the LGBM hyperparameters. All training was carried out with 9-fold cross validation.

## 4 EXPERIMENTS

### 4.1 Experimental Setting

To evaluate the effectiveness of our MACE predictive model, Our (CA), a comparative analysis was conducted with the three benchmark models: ASCVD, LGBM, and Our (SA). ASCVD is an existing MACE prediction model that relies solely on health checkup data, while our models and the LGBM make predictions according to both health checkup data and medical claims data. When using only health checkup data, our model generates predictions directly through the FFN block to the classification head. Our (SA) was developed by us based on self-attention mechanisms with reference to previous studies [13, 21, 22, 24]. In the experiments, the data were divided into a ratio of 8:1:1 for the training, validation, and test sets, using Stratified K-Fold for the splitting. The average score was evaluated for nine models trained using data from 9-fold cross-validation (the metrics are described in Section 4.3). Furthermore, the Wilcoxon rank-sum test was conducted on our proposed model and the best-performing model among the others, verifying that the superior performance of our proposed model is statistically significant. The DeLong test [7] was also performed on the ensemble of prediction results from the nine models, and the details are presented in Appendix H.

### 4.2 Datasets

*4.2.1 Data Source.* Three types of data were obtained from the Health Insurance Association for Architecture and Civil Engineering Companies in Japan: 1) health checkup data for the 2014 fiscal year comprising data from 166,030 individuals; 2) medical claims data from May 2014–April 2022 covering 714,710 individuals; and 3) updated insurance qualification data as of April 2022 for 1,484,255 individuals. Table E.7 provides an overview of the data sources.

*4.2.2 Creation of the Dataset.* To include individuals with sufficient MACE observation periods and those whose MACE could be predicted using both benchmark models and our proposed model, we selected 51,367 experimental subjects according to the flow chart shown in Figure G.5. The prediction task, referring to the occurrence of MACE, was created using the medical information in the claims data spanning April 2015–March 2022. Table F.8 provides an overview of the final dataset. The group of 51,367 experimental subjects was divided for the training and evaluation processes, designating 41,094 to training, 5,136 to validation, and 5,137 to testing. The incidence of hospitalization or death due to MACE among the experimental subjects is detailed in Table 2. The input features were

created from the health checkup data and claims data for the period April 2014–March 2015.

**Table 2: Incidence of MACE and its components (n=51,367)**

| MACE | AMI | CEREBRO | HF | PAD |
|------|-----|---------|-----|-----|
| 1,223 (2.38%) | 669 (1.30%) | 486 (0.95%) | 93 (0.18%) | 20 (0.04%) |

The expanded forms of the diseases comprising MACE, as listed in the table, are acute myocardial infarction (AMI), cerebrovascular disease (CEREBRO), heart failure (HF), and peripheral arterial disease (PAD).

### 4.3 Metrics

The performance of the MACE prediction models was evaluated using ROC-AUC and the Matthews correlation coefficient (MCC) [2]. The MCC was calculated at the 0.5 threshold, defining the occurrence of MACE.

## 5 RESULTS

### 5.1 Experiment Results

The models that utilized both health checkup data and medical claims data outperformed the ASCVD model that only used health checkup data (Table 3). Furthermore, when we adopted a cross-attention mechanism in our model, it outperformed the others, giving the highest scores, with an ROC-AUC of 0.7720 and an MCC of 0.1525. For the two best-performing models, we conducted the Wilcoxon rank-sum test using the 9 ROC-AUC values and 9 MCC values measured. This test confirmed statistical significance (p-values = 0.0039, 0.0039). We also conducted the DeLong test for these models, with the results in Table H.9.

**Table 3: Results of MACE prediction (Mean and Std)**

| Model | Input[1] | Metrics | |
|-------|----------|---------|---|
| | | ROC-AUC | MCC[2] |
| Our (FFN) | HC | 0.7412 (0.0086) | 0.1167 (0.0044) |
| Our (SA[3]) | HC+MC | 0.7586 (0.0062) | 0.1253 (0.0069) |
| Our (CA[4]) | HC+MC | **0.7720 (0.0059)** | **0.1525 (0.0089)** |
| ASCVD | HC | 0.7521 | 0.0891 |
| LGBM | HC | 0.7374 (0.0134) | 0.1202 (0.0076) |
| | HC+MC | 0.7559 (0.0147) | 0.1256 (0.0060) |

The ASCVD model, already trained, was directly applied to the test data. Our (SA)[3] model does not utilize the cross-attention block applied in our proposed model architecture (See Figure 1).

1. HC means health checkup data and MC means medical claims data.
2. MCC is evaluated at the 0.5 threshold.
3. SA means self-attention mechanism.
4. CA means cross-attention mechanism.

## 6 DISCUSSION & CONCLUSION

MACE prediction performance was enhanced by applying a cross-attention mechanism to clinically unstructured medical claims data. Our proposed method effectively weights and learns the many-to-many relationships between diagnoses and treatments, effectively

utilizing large-scale and clinically unstructured data. Applying our proposed model to identify individuals at high-risk of MACE has the potential to improve public health services in Japan (details described in Appendix I). In the future, we plan to use the cross-attention mechanism for pre-training on a large volume of medical claims data, aiming to develop pre-training models that are more effective for the prediction of various diseases (downstream tasks).

## 7 LIMITATIONS

This study has excluded data that may lead to improved prediction performance, including that from duplicated medical codes and time-series information from medical claims data. Additionally, the findings of this study should not be extrapolated to populations with a very high number of medical codes recorded in a single month, as they have been excluded from the experimental dataset. Further details regarding the study limitations are presented in Appendix J.

## 8 ACKNOWLEDGMENTS

We thank the Health Insurance Association for Architecture and Civil Engineering Companies for their support in developing the database.

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

# APPENDIX

## A   MODEL ARCHITECTURE SUPPLEMENT

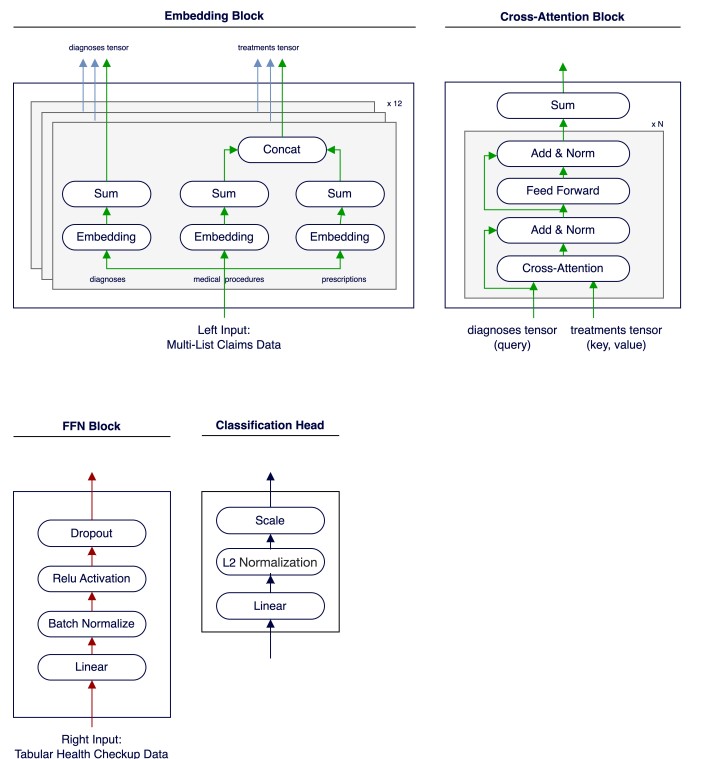

Figure A.3: Blocks in our models. Additional details on the embedding block are provided in Figure A.4, while further explanations on the cross-attention block can be found in Figure 2. The cross-attention block is adopted in the proposed model, not in our model based on the self-attention mechanism (Our SA).

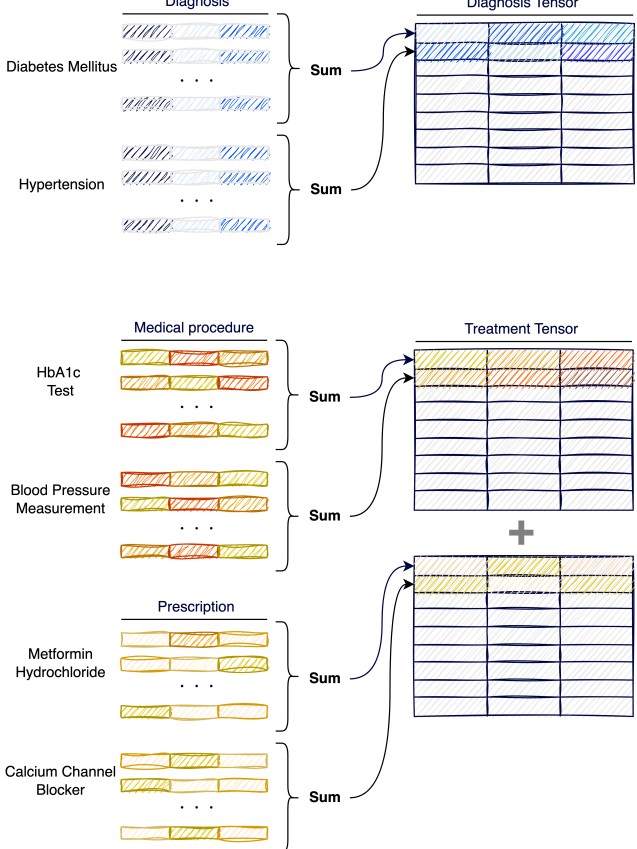

Figure A.4: Flow diagram illustrating the embedding block. During the diagnostic process, sub-tokens are initially transformed into vectors, which are then aggregated to generate a singular "diagnosis" vector. Similarly, sub-tokens related to medical procedures and prescriptions undergo embedding into vectors and, following aggregation, form "medical procedure" and "prescription" vectors, respectively. These two vectors are subsequently concatenated to represent treatment information. It is essential to note that when dealing with a month's worth of medical claims data, encompassing numerous diagnoses, medical procedures, and prescriptions, both diagnostic and treatment information is presented as tensors. These tensors possess dimensions of batch size, length, embedding size, with the length typically spanning several dozen.

## B HYPERPARAMETER SUPPLEMENT

**Table B.4: Hyperparameters of our proposed model**

| Category | Parameter | Value |
|---|---|---|
| Batch | batch_size | 2,048 |
| Embedding block | num_embeddings | 20,000 |
| | embedding_dim | 128 |
| | dropout | 0.01 |
| Transformer encoder | d_model | 128 |
| | nheads | 4 |
| | dim_feedforward | 64 |
| | dropout | 0.01 |
| | num_layers | 3 |
| Cross-Attention | d_model | 128 |
| | nheads | 4 |
| | dim_feedforward | 64 |
| | dropout | 0.01 |
| | num_layers | 2 |
| FFN block | in_features | 8 |
| | out_features | 128 |
| | dropout | 0.01 |
| Classification head | in_features | 256 |
| | out_features | 2 |
| Loss function | name | Class balanced focal loss |
| | beta | 0.99999 |
| | gamma | 2.0 |
| Optimizer | name | AdamW |
| | lr | 5e-04 |
| | beta1 | 0.9 |
| | beta2 | 0.999 |
| | eps | 1e-08 |
| | weight_decay | 1e-04 |

## C HEALTH CHECKUP DATA SUPPLEMENT

**Table C.5: Definition of input data from health checkup data.**

| Features | Definition | Used columns |
|---|---|---|
| Male | Male: 1, female: 0 | Sex |
| Age | Age as of March 31, 2015, based on date of birth. | Date of birth |
| Systolic blood pressure | Last measurements are used. | Systolic blood pressure measurement 1st, 2nd, and 3rd |
| Total cholesterol | Calculated using Friedewald's formula when triglyceride < 400 mg/dL. If triglyceride $\geq$ 400 mg/dL, we treat total cholesterol as a missing value. | Triglyceride, HDL cholesterol, LDL cholesterol |
| HDL cholesterol | HDL cholesterol value | HDL cholesterol |
| Diabetes | Diabetes is defined when any of the following conditions are met:
• HbA1c $\geq$ 6.5%
• Fasting glucose $\geq$ 126mg/dL
• Causal glucose $\geq$ 200mg/dL
• Diabetes medication

Exceptionally, we treat the variable diabetes as a missing value.
• Diabetes medication are not self-reported.
• Any HbA1c and blood glucose were not measured. | Diabetes medication, HbA1c, fasting glucose, causal glucose |
| Antihypertensive medication | Taking: 1; not taking: 0 | Antihypertensive medication |
| Smoking | Smoking: 1, not smoking: 0 | Smoking |

We processed the values stored in the columns of health checkup data to create features based on the definitions provided.
However, for male, antihypertensive medication, and smoking, we used the values stored in the health checkup data directly as features.

# D    CONVERSION METHOD FOR MEDICAL CODE SUPPLEMENT

**Table D.6: Method of converting from medical codes to sub-tokens.**

| Medical code | Conversion Method |
| --- | --- |
| Diagnosis code | We referred to the five categories detailed in the WHO ICD-10 Code Instruction Manual (2.4.2 - 2.4.6), namely: Chapters, Blocks of Categories, Three-Character Categories, Four-Character Subcategories, and Supplementary Subdivisions for Use at the Fifth or Subsequent Character Level. We have developed a method to convert each diagnosis code (ICD-10 code) recorded in medical claims data into a list of sub-tokens corresponding to these five categories. |
| Medical procedure code | Medical procedure code recorded in medical claims data are composed of five categories (chapter, section, division, branch, item) and are represented as a ten-digit number. For example, the medical procedure code for HbA1c test (numbered as 2030050900) is composed of the following five categories: 2 (specific medical service fee) for the chapter, 03 (test) for the section, 005 (test of hematology) for the division, 09 (Hemoglobin A1c) for the branch , 00 (not applicable) for the item. 

 We have developed a method to combine these five categories into a list of sub-tokens, each containing five elements. The elements of this sub-token list sequentially represent the chapter alone, from chapter to section, from chapter to division, from chapter to branch, and from chapter to item (equivalent to the medical procedure code). Therefore, the medical procedure code for "HbA1c test" can be converted into the following sub-token list: [2, 203, 203005, 20300509, 2030050900]. 

 As higher order categories in the medical procedure code provide more significant medical information, we created a sub-token list combining these categories. For example, the medical significance of the division code "001" varies depending on its higher order categories, such as chapter and section. If the combined chapter and section number is "201", followed by the division number "001", the code relates to the medical supervision of a specific disease. However, if the combined chapter and section number is "203", with the same division number "001", the code then pertains to a urinary test. |
| Prescription code | Prescription (prescribed medication) code recorded in medical claims data are composed of three categories (medication effect, administration routes and ingredients, and medication shape) and are represented as an 8-digit alphanumeric code. 

 For example, the prescription code for Metformin Hydrochloride, an antidiabetic medicine, is 3962002F. This code comprises the following three categories: 3962 (medication effect), 002 (administration routes and ingredients), and F (medication shape). Similar to the medical procedure codes, we combined these categories to create a list of sub-tokens consisting of three elements. Therefore, the prescription code for Metformin Hydrochloride "3962002F" is converted into the following sub-token list: [3962, 3962002, 3962002F]. |

## E  SUMMARY OF DATASOURCE SUPPLEMENT

**Table E.7: Summary of datasource**

|  | Health checkup data | Medical claims data[1] | Qualification of insurance data[2] |
|---|---|---|---|
| Number of records | 179,309 | 20,317,245 | 1,487,419 |
| Number of individuals | 166,030 | 714,710 | 1,484,255 |
| Period | 2014/4 – 2015/3 | 2011/5 – 2022/3 | 1943/4 – 2022/4 |
| Coverage rate of prefectures with medical institutions | - | 100% | - |

1. Medical claims data reviewed from May 2014 to April 2022 are used. Due to delays in the review of medical claims data, these data include medical services conducted before April 2014.

2. These data include people who have lost their insurance. The number of insured individuals was 398,239 as of 31 March 2022.

## F  SUMMARY OF DATASET SUPPLEMENT

**Table F.8: Summary of dataset**

| Variables (unit) | Datasource | | Value | |
|---|---|---|---|---|
|  | Health checkup data | Medical claims data | Mean[1] | STD[2] |
| Age[3] (years) | ✓ |  | 47.8 | 6.3 |
| Male (binary) | ✓ |  | 73.1 | - |
| Systolic blood pressure (mmHg) | ✓ |  | 122.9 | 16.2 |
| Total cholesterol (mg/dL) | ✓ |  | 212.7 | 35.4 |
| HDL cholesterol (mg/dL) | ✓ |  | 62.5 | 16.9 |
| Diabetes (binary) | ✓ |  | 7.1 | - |
| Antihypertensive medication (binary) | ✓ |  | 14.0 | - |
| Smoking (binary) | ✓ |  | 28.3 | - |
| The number of invoiced months in medical claims data per individual (months) |  | ✓ | 4.9 | 3.6 |
| The number of unique medical tokens per individuals (token counts) |  | ✓ | 32.4 | 21.9 |

The dataset includes 7,031 unique medical tokens and 12,068 unique medical sub-tokens extracted from medical claims data.

1. The unit for mean of binary variables is percentage, while the unit for mean of the other variables corresponds to the unit of the variable itself.

2. The standard deviation (STD) was calculated for all variables excluding binary variables.

3. Age was calculated as of 31 March 2015.

# G  SELECTION PROCESS SUPPLEMENT

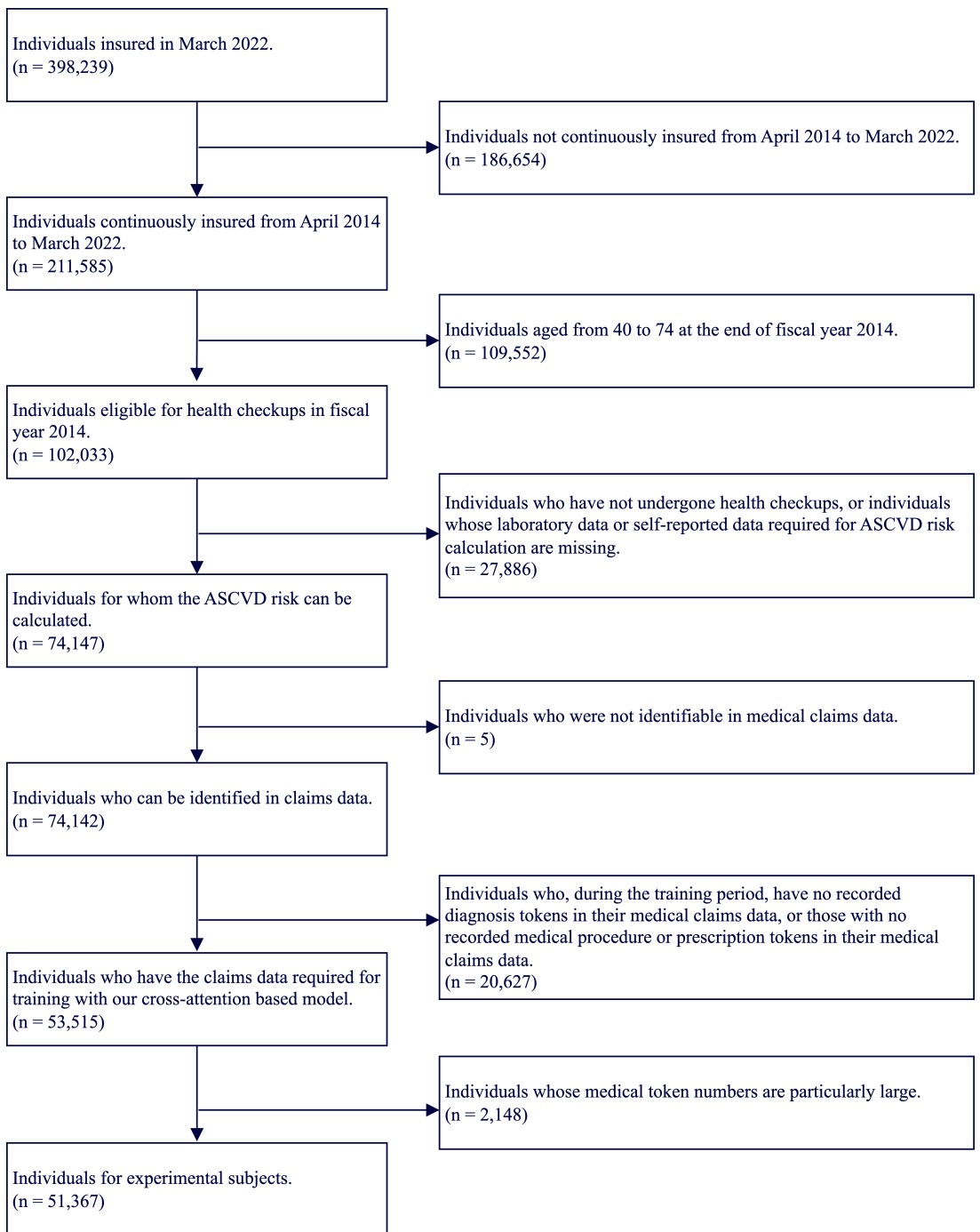

**Figure G.5: Selection process for experimental subjects.**

## H ADDITIONAL EXPERIMENTAL RESULTS SUPPLEMENT

The prediction results of nine models created using nine-fold cross-validation were averaged to construct an ensemble model. The ROC-AUC scores for the ensemble models of the two best-performing models were 0.7791 and 0.7653 for Our (CA) and Our (SA), respectively. The MCC scores were 0.1571 and 0.1368 for Our (CA) and Our (SA), respectively. The prediction performance of Our (CA) was superior to that of Our (SA). Additionally, a significance test for the ROC-AUC (Delong test) yielded a p-value of 0.0159, indicating statistical significance.

**Table H.9: Results of MACE prediction by ensemble models**

| Model | Input[1] | Metrics | |
| | | ROC-AUC | MCC[2] |
| --- | --- | --- | --- |
| Our (SA[3]) | HC+MC | 0.7653 | 0.1368 |
| Our (CA[4]) | HC+MC | **0.7791** | **0.1571** |

1. HC means health checkup data and MC means medical claims data.
2. MCC is evaluated at the 0.5 threshold.
3. SA means self-attention mechanism.
4. CA means cross-attention mechanism.

## I USE CASES SUPPLEMENT

Our proposed model for identifying high-risk individuals for MACE can be effectively applied in the medical and community health fields in Japan. By adjusting the threshold of the prediction probabilities, the model can be tailored to specific interventions such as those targeting many people with moderate or higher risk, those for a small group with high risk, and those conducted in order of highest risk according to available resources. We developed this model to improve the public health services in Japan. Identifying high-risk individuals for MACE and providing appropriate interventions is an important public health strategy in Japan, where a national screening program has been in place since 2008. However, existing models that rely solely on health checkup results have been inadequate in capturing many risk factors. The proposed model, which utilizes medical claims data and employs cross-attention mechanism, enables more accurate identification of high-risk individuals.

## J LIMITATION SUPPLEMENT

In this study, due to computational resource limitations, we trained using input features with duplicates removed from the monthly diagnosis and treatment history, an approach also adopted in a previous study [24]. However, repeated diagnoses and treatments can be a significant source of information. Thus, future research will consider ways to incorporate duplicate information without compromising computational efficiency.

We have established exclusion criteria when selecting the subjects for our experiments. For example, individuals with numerous medical codes even after removing duplicates were excluded. Building a highly general model suitable for such special cases is somewhat challenging. Therefore, we initially proceeded with experiments on general cases. However, models applicable to special cases are important, and we plan to address this as a future research task.

We did not use monthly information in our proposed model because our experiments using positional encoding of monthly data did not result in performance improvements. However, time-series information is important, and we believe that its effective use could enhance performance.