# OpenReview forum: "In Medical Claims Data, Enhancing Predictive Performance for Major Adverse Cardiovascular Events Using Cross Attention"
_KDD.org/2024/Workshop/AIDSH — KDD-AIDSH 2024 Oral_

### Official Review · Reviewer_uKue · 2024-06-11
**Review #4**

**Rating:** 5
**Confidence:** 4

**Review:**

Note: The ACKNOWLEDGMENTS section disobeys the double-blind rule

**Summary:**
The paper enhances MACE prediction using medical claims data through a cross-attention mechanism, achieving a ROC-AUC score of 0.7720, surpassing traditional models like ASCVD and LGBM. By effectively weighing diagnoses and treatments, it improves the identification of high-risk individuals, indicating potential benefits for public health and broader applications in healthcare research.

**Pros:**
- Clear writing and presentation
- Sound methodology design (cross-attention for unstructured medical claims data's diagnoses and treatments characteristics)
- The impact of the problem and data is significant (Medical Claims Data, Major Adverse Cardiovascular Events)

**Cons:**
- Limited evaluation with only one private data source
- It is better to include more metrics apart from AUROC and MCC. E.g. AUPRC, F1
- The architecture design lacks novelty; the cross-attention mechanism was introduced at least 3 years ago.
- Limited baseline models are compared. E.g. those multimodal electronic health record data modeling works. E.g.
    - [2023-ICML] Improving Medical Predictions by Irregular Multimodal Electronic Health Records Modeling
    - [2023-IJCAI] VecoCare Visit Sequences-Clinical Notes Joint Learning for Diagnosis Prediction in Healthcare Data

---

### Official Review · Reviewer_dRDg · 2024-06-14
**Review comments**

**Rating:** 7
**Confidence:** 4

**Review:**

In this work, the authors propose a cross-attention model to predict major adverse cardiovascular events (MACE) using medical claims data. The paper is well-written, and the structure is clear. The research question may be clinically meaningful. My major comments are:

1. Can the proposed model process temporal information in patients' records, or is it limited to handling tabular data?
2. Given the high imbalance of the labels, AUPRC would be a more informative metric to evaluate model performance.
3. I suggest that the authors add more baselines, such as a naive transformer model. It is unclear whether the current performance improvement is due to other components such as focal loss.

---

### Decision · Program_Chairs · 2024-06-28

Accept (Oral)